# The Impact of Mindful Learning on Subjective and Psychological Well-Being in Postgraduate Students

**DOI:** 10.3390/bs13121009

**Published:** 2023-12-10

**Authors:** Qing Wang, Yuanyuan Zhang, Ying Zhang, Tingwei Chen

**Affiliations:** Shanghai Key Laboratory of Mental Health and Psychological Crisis Intervention, School of Psychology and Cognitive Science, East China Normal University, Shanghai 200062, China; yuanyuan.zhang@uconn.edu (Y.Z.); yingzhang2021@zju.edu.cn (Y.Z.); 51183200059@stu.ecnu.edu.cn (T.C.)

**Keywords:** mindful learning, subjective well-being, psychological well-being, coaching, postgraduate students

## Abstract

Mindful learning is widely known to improve learning outcomes, yet its association with students’ well-being remains unexplored. This study aimed to investigate the impact of mindful learning on subjective well-being (SWB) and psychological well-being (PWB) in postgraduate students, using survey questionnaires and a randomized experimental design. In Study 1, correlation and regression analyses based on 236 postgraduate students revealed significant positive associations among mindful learning, SWB, and PWB. In Study 2, 54 students were randomly assigned to three groups: the experimental (which received Mindful Learning Coaching), active-, and blank control groups. The results from repeated-measures ANOVA showed that coaching significantly improved students’ mindful learning. The participants’ SWB and PWB significantly decreased in both the active- and blank control groups, whilst their SWB and PWB tended to increase in the experimental group. In conclusion, mindful learning, SWB, and PWB are significantly correlated, while the enhancement of mindful learning may be a protective factor in students’ well-being.

## 1. Introduction

Postgraduate students’ learning and psychological conditions are primary concerns in higher education [1,2,3,4]. An effective learner is equipped with learning capabilities: the abilities, skills, and strategies that learning requires [5,6]; non-cognitive factors such as personal qualities, grit, and mindset [7]; and mindful learning—the learner has the ability to be aware, perceive, and conceive during their learning process [8,9]. When learning is mindful, learners are sensitive to context, open to new information, aware of novel distinctions and, eventually, can develop multiple perspectives [10]. Mindful learners are more persistent and engaged in learning [11] and have positive academic achievements [12,13]. Indeed, mindful learning correlates with several cognitive and affective outcomes, such as students’ cognitive flexibility [14], competency [15], academic self-efficacy [16], and academic emotions [17]. These cognitive and affective factors in learning play important roles in students’ academic life, including their psychological health [18]. However, few studies have directly investigated the relationship between mindful learning and students’ well-being. In the current study, we utilized survey questionnaires and a randomized experiment to investigate the impact of mindful learning on postgraduate students’ subjective and psychological well-being. 

### 1.1. Mindful Learning

Mindful learning, as applied in this study, is defined as the application of a flexible state of mind—characterized by active engagement in the present, awareness of learning context, sensitivity to novelty, and the creation of new definitions and categories of information [19]—to the learning process. This concept draws on Langer’s [19,20] definition of mindfulness, which emphasizes being present and aware, paying attention to the current moment, openness to new possibilities, liberation from automatic and habitual ways of thinking and behaving, and fostering creativity and innovation. Following Langer [10,20], mindful learning involves approaching learning with an open, curious, and non-judgmental attitude, thus avoiding “autopilot” engagement. It indicates a learning process that emphasizes calmness, sensitivity, and openness in thinking—individuals are situated in the present and deeply immersed in learning, actively draw distinctions and notice new things, and see the familiar in the novel and the novel in the familiar [10]. Mindless learning, as the opposite, indicates relying on previously established categories and premature cognitive commitment. When people are mindless in learning, they behave according to categories created in the past, are entangled in a single and inflexible perspective, and are unaware of other possible ways of knowing [21].

The number of empirical studies on mindful learning in educational settings is rising. For instance, mindful learning is associated with generating new thoughts, cultivating intelligence [13], and developing meta-awareness of the learning process [22]. Mindful learning has been found to be positively associated with creativity [23,24,25] and critical thinking [22], while negatively correlated with students’ classroom negative behaviors [26]. Moreover, mindful learning is a powerful predictor of positive learning experiences (e.g., mastery experience through flow and self-efficacy) and leads to better learning outcomes [10,16]. However, there have been few empirical studies on the direct impact of mindful learning on students’ well-being or psychological health, which is the aim of the current study.

### 1.2. The Relationship between Mindful Learning and Well-Being

Well-being is a multidimensional construct that includes two empirically distinct concepts: subjective well-being (SWB) and psychological well-being (PWB) [27,28]. SWB refers to a general evaluation of quality of life according to individuals’ own criteria, which comprises the three dimensions of life satisfaction, relatively high positive affect, and relatively low negative affect [29,30,31,32]. The concept of PWB emphasizes the potential for self-actualization, self-growth, and positive experience that is obtained by individuals in their efforts to engage in life activities, and for achievement [33,34,35,36]. PWB comprises six dimensions: autonomy, environment mastery, self-acceptance, positive relationships with others, personal growth, and purpose in life [34,35]. Although SWB and PWB are derived from separate philosophical traditions (i.e., hedonism and eudaimonism) and have different measures [37], some of their dimensions are empirically linked [27,35,36,38,39]. 

In educational settings, students’ well-being can be affected by various elements that are related to learning, e.g., academic achievement, self-esteem, self-efficacy, personality traits, achievement motivation, emotional intelligence, and attributional styles [3,40,41,42,43,44]. These elements are largely related to attentional, emotional, and motivational aspects of mindful learning that can be addressed by the mindfulness theory and the self-determination theory (SDT).

Drawing from the mindfulness theory, mindfulness has been shown to enhance both psychological and subjective well-being. Therefore, when this mindful approach is applied to learning, forming what we term as mindful learning, it is expected to boost students’ well-being. Mindful learning emphasizes learners’ implicit awareness of their learning environments, non-judgmental view of experiences, continuous creation of new categories, and openness to new information and diverse perspectives, which helps them make more effective learning decisions and strategies [9,19]. 

Although we primarily refer to Langer’s [10] theory of mindfulness to define mindful learning in this study, we recognize the fundamental contribution of Kabat-Zinn’s [45] definition of mindfulness in promoting attention to the present moment in an accepting and non-judgmental way. It is crucial to distinguish and overlap the features of these two approaches. Langer’s approach to mindfulness is characterized by the process of actively creating novel distinctions about a situation and its environment, fostering multiple perspective-taking, while Kabat-Zinn’s theory emphasizes purposeful attention to the present moment in a non-judgmental manner [46]. Both theories, although seemingly different, agree on mindfulness as a state of being present and aware. Hence, mindful learning, as per our definition, represents an amalgamation of these two theories, stressing active engagement, awareness, and non-judgmental acceptance during the learning process. Mindfulness has been found to be correlated with a deeper level of learning engagement [47]. Mindfulness-based training in schools can enhance flexibility in thinking and openness to experience [48], as well as reduce stress and increase academic achievement [49,50]. The relationship between mindfulness and well-being has been extensively studied [51]. Mindfulness has been found to be positively correlated with PWB [52,53], while mindfulness-based training can enhance individuals’ PWB by attenuating reactivity to emotional stimuli [54]. Moreover, mindfulness has a significant positive correlation with SWB; specifically, people with higher trait mindfulness tend to experience higher levels of life satisfaction and positive affect, and lower levels of negative affect [55,56]. Mindfulness-based intervention can significantly improve individuals’ SWB and reduce anxiety and stress [57,58,59], while it effectively increases students’ emotional management ability when they face challenges in learning [60,61].

In addition, we propose that self-determination theory (SDT) plays a distinctive role in the potential impact of mindful learning on students’ SWB and PWB, particularly autonomous motivation and needs satisfaction. Mindful learning, by fostering awareness, non-judgment, openness, and flexibility, can facilitate the satisfaction of these needs. In turn, the satisfaction of these needs is associated with enhanced well-being, bridging the connection between Mindful Learning, SDT, and well-being. SDT holds the perspective that people have the potential and tendency for self-realization and independently choose activities that contribute to self-growth, considering their own needs and the external social environment [62,63]. Autonomous motivation indicates a high degree of self-determination, which is regarded as an important source of learners’ autonomy [64]. From the view of SDT, three basic needs (i.e., autonomy, relatedness, and competence) are the main components of individuals’ optimal functioning and PWB [39]. The importance of relatedness for both learning and well-being is a central concept in mindful learning [24,25]. As learners engage mindfully with their learning environments, they cultivate a greater sense of relatedness, which contributes to improved well-being. Studies have shown that individuals with high self-determination tend to demonstrate stronger autonomous motivation, hold intrinsic goals and aspirations, and achieve higher levels of PWB [65,66]. Moreover, empirical studies on SWB suggest that satisfaction of basic needs increases the level of sport self-determination and predicts SWB in athletes [67]. Compared with controlled motivation, a higher level of autonomous motivation predicts higher levels of SWB [68], life satisfaction, and positive affect, and a lower level of negative affect [69,70]. 

Based on the literature, we designed a cross-sectional study to explore the direct relationships between mindful learning, SWB, and PWB using survey questionnaires. The hypotheses in Study 1 are as follows: 

**H1.** 
*Mindful learning is significantly associated with SWB and PWB.*


**H2.** 
*Mindful learning can positively predict SWB and PWB.*


To investigate the impact of mindful learning on students’ well-being, we determine whether the development of mindful learning can lead to changes in SWB and PWB. Mindful learning can be enhanced through a coaching psychology approach [71,72]. Coaching psychology aims to promote well-being, performance, learning, and growth in individuals who are not clinically diagnosed as having severe mental health concerns [73,74,75]. Coaching, as a facilitative methodology, has shown its effectiveness in developing students’ cognitive, affective, and behavioral aspects [76,77,78,79,80]. 

In Study 2, we examined whether mindful learning can impact the development of postgraduate students’ SWB and PWB by employing a randomized controlled trial (RCT) experimental design. We facilitated students’ development of mindful learning via the Mindful Learning Coaching (MLC) program, which was designed based on the original framework of Mindful Agency Coaching [81,82]. The MLC program integrates narrative, positive psychology, and mindfulness approaches (see Methods). The narrative approach helps individuals develop positive personal narrations and positive self-identities that enhance a sense of ownership and responsibility in learning [83,84]. The positive psychology approach emphasizes positive emotions and thoughts [85] as well as the exploration of students’ signature strengths, core values, available resources, and inner motivations in learning [86,87]. Last, the mindfulness approach focuses on cultivating self-awareness, reflectiveness, and a non-judgmental attitude through mindfulness practices such as mindfulness-based stress reduction (MBSR) [88] and mindfulness-based cognitive therapy (MBCT) [89]. The hypotheses in Study 2 are as follows:

**H3.** 
*Following the MLC intervention, the participants in the experimental group significantly increase their SWB and PWB compared with those in the active- and blank control groups.*


**H4.** 
*An increase in mindful learning leads to increases in SWB and PWB for participants in the experimental group.*


## 2. Study 1

### 2.1. Design and Participants

Study 1 was a cross-sectional study that investigated the relationships between mindful learning, SWB, and PWB in postgraduate students. The inclusion criteria for participants were individuals aged between 18 and 60, native Mandarin speakers, and current postgraduate students. G-power was used to estimate the appropriate sample size [90]. To test the linear multiple regression, *n* = 246 was an acceptable sample size under the conditions of effect size, *ρ^2^* = 0.06 (medium effect size), POWER, (1 − *β*) = 0.80, α = 0.05, and number of predictors = 8. A total of 254 students were initially recruited by an online crowdsourcing platform by convenience sampling method. The study was approved by the University Committee on Human Research Protection, which is affiliated to the institution. All the participants signed informed-consent forms and completed the online questionnaires voluntarily. After screening for the validity and authenticity of the participants’ responses (valid response rate = 92.91%), the final sample included 236 participants (age mean = 23.72, *SD* = 1.97, range from 20 to 34) with 80 males (33.90%) and 156 females (66.10%). Of the participants, 44.10% had had meditation experience (including mindfulness, vipassana, yoga, tai chi, or Buddhism-based meditation). Using independent samples *t*-tests, we tested for gender differences in each variable as well as differences in the meditation experience. A significant difference in terms of gender was observed in autonomy (*p* < 0.01), a subscale of PWB, and positive affect (*p* < 0.05). The participants who had meditation experience obtained higher positive affect scores (*p* < 0.01). Age and positive affect were significantly correlated (*r* = 0.13, *p* = 0.041). The participants’ demographics are shown in Table 1. 

### 2.2. Measures and Analysis

Mindful learning was measured by the Chinese-validated version of the Mindful Agency Scale [64,91,92]. Mindful agency indicates a strong sense of autonomy, intentionality, and self-awareness in the observation, management, and regulation of the motivational, cognitive, emotional, and social processes in learning [64,92]. There are five dimensions of mindful agency: (1) Learning methods: learners adapt and flexibly adopt different methods in various learning activities; (2) Emotional regulation: learners manage negative emotions when facing obstacles, thereby motivating themselves to learn; (3) Awareness of planning: learners are cognizant of their learning goals and strategize pathways to achieve these goals in specific learning tasks; (4) Learning engagement: learners actively participate in and deeply immerse themselves in learning activities, experiencing a state of flow; and (5) Openness to experiences: learners maintain an open stance towards diverse learning experiences, acknowledging them as an integral part of the learning process. This scale was considered appropriate for measuring mindful learning because mindful agency indicates an individual’s disposition towards embracing mindfulness during learning (mindful learning). The scale comprises 16 items answered on a 6-point Likert scale ranging from 1 (very inconsistent with me) to 6 (very consistent with me), e.g., “When I feel frustrated with my studies, I am very good at finding ways to make myself find the feeling of learning”. 

Subjective well-being was measured using the Chinese version of the Subjective Well-Being Scale [93], which is widely used in Chinese and is reported to have good validity and reliability [94,95]. The scale comprises three subscales: life satisfaction, positive affect, and negative affect. The life satisfaction subscale comprises five items rated on a 7-point Likert scale ranging from 1 (strongly disagree) to 7 (strongly agree), e.g., “In most ways, my life is close to my ideal.” The average score across the five items represents the score for life satisfaction [96]. The positive and negative affect subscales comprise six and eight adjectives, respectively, with each word describing a positive (e.g., proud) or a negative (e.g., angry) emotion. The participants were asked to respond on how often they experienced these emotional states in the last week of the study, on a 7-point scale ranging from 1 (not at all) to 7 (all the time) [97]. The average frequencies of the positive and negative affect were calculated as the frequencies of the experienced positive and negative affect. 

PWB was measured using Ryff’s short measurement of PWB [98]. This instrument was translated from English to Chinese and back-translated by the researchers based on previous Chinese versions [99]. The scale comprises 18 items (three for each construct) answered on a 6-point Likert scale ranging from 1 (strongly disagree) to 6 (strongly agree), e.g., “For me, life has been a continuous process of learning, changing, and growth.” The total PWB score was computed by summing up the scores for all the 18 items.

The reliability of the measures and subscales is reported in Table 1. Considering that the reliability of the PWB subscales was below 0.60, we only used the total scores of PWB in the follow-up analysis. The software, IBM SPSS 23.0 Statistics for Windows, was used for descriptive, correlation, and regression analyses. 

### 2.3. Results

#### 2.3.1. Descriptive and Correlation Analyses

The means and standard deviations of all the measures and their inter-correlations are shown in Table 2. Pearson bivariate correlation coefficients were calculated for each measure. As expected, mindful learning was positively associated with PWB (*r* = 0.49, *p* < 0.001). For SWB, mindful learning was positively correlated with life satisfaction (*r* = 0.40, *p* < 0.001) and positive affect (*r* = 0.42, *p* < 0.001), while it was negatively correlated with negative affect (*r* = −0.28, *p* < 0.001). The results support the hypothesis (H1) that mindful learning is significantly associated with SWB and PWB.

#### 2.3.2. Regression Analysis

The results of the linear regression analysis showed that the participants’ mindful learning positively predicted PWB (*β* = 0.484, *p* < 0.001) (See Table 3a). The total variances explained by the model for PWB were 22.7% [*F* (4, 231) = 18.217, *p* < 0.001]. Standard multiple regression analyses were used to determine the effects of specific dimensions of mindful learning on SWB and PWB. For PWB, we found that learning methods (*β* = 0.313, *p* < 0.001) had a significant positive effect on PWB, whereas the other four dimensions had no significant effect. Together, the five dimensions explained 24.6% [*F* (8, 227) = 10.570, *p* < 0.001] of the variance in the dependent variable (PWB). 

For SWB, we found that mindful learning has a significant positive impact on both life satisfaction (*β* = 0.396, *p* < 0.001) and positive affect (*β* = 0.410, *p* < 0.001), as well as a significant negative impact on negative affect (*β* = −0.275, *p* < 0.001) (See Table 3b). Further detailed analysis revealed that emotion regulation (*β* = 0.280, *p* < 0.001) exerted a significant positive effect on life satisfaction, while learning methods were shown to significantly enhance positive affect (*β* = 0.195, *p* < 0.01) and significantly reduce negative affect (*β* = −0.217, *p* < 0.001). The results support the hypothesis (H2) that mindful learning positively predicts students’ SWB and PWB.

## 3. Study 2

### 3.1. Design and Participants

Study 2 was an RCT study to further examine the impact of mindful learning on students’ SWB and PWB through the MLC intervention. We use G-power to test the appropriate sample size of interaction of repeated-measurement ANOVA. Under the conditions of effect size, *f* = 0.25 (medium effect size), POWER, (1 − *β*) = 0.80, and α = 0.05, *n* = 42 was an acceptable sample size.

The participants in Study 2 were independently recruited and different from those in Study 1. Fifty-four postgraduate students were recruited from a Chinese university via an online advertisement, posters, websites, and text messages. The study protocol was approved by the University Committee on Human Research Protection, which is affiliated to the institution. All the participants in the study volunteered and signed informed-consent forms. After the recruitment of participants, a research student who was blind to the study design numbered the participants (number 1–54), and these numbers were randomly grouped into three groups using a random grouping tool (RANDOM.ORG). The participants were randomly assigned into three groups: (a) the experimental group (*n* = 18), in which the participants attended MLC; (b) the active control group (*n* = 18), in which the participants joined a relaxation program; and (c) the blank control group (*n* = 18), in which the participants did not receive any form of intervention. The flow of the participants and the study procedure are shown in Figure 1. 

All the participants were asked to fill in the questionnaires before and after the intervention. Three participants in the experimental group and three participants in the active control group withdrew due to time conflicts. The final sample comprised 48 participants, ranging in age from 21 to 30 years (*M* = 23.94, *SD* = 2.04), with 15 in the experimental, 15 in the active control, and 18 in the blank control groups. There was no statistical difference in gender or age among the three groups in the baseline case (see Table 4). 

### 3.2. Measures

The measurements for mindful learning, SWB, and PWB were the same as the scales used in Study 1. The reliability of these scales in the pre- and post-tests is shown in Table 5. 

### 3.3. Intervention Procedure

The participants in the experimental group were invited to a four-week group-based MLC from December 2019 to January 2020. The MLC facilitator was one of the authors, who had been learning and practicing mindfulness and coaching for two years. The current MLC design was based on the original framework of Mindful Agency Coaching that can effectively develop mindful agency [81,82], with substantial adaptations for the postgraduate students in this study. The MLC comprised four one-hour workshops, one per week (see Table 6 for detailed schedules, activities, and aims). During the initial two weeks, our sessions followed a 20–20–20 structure: 20 min for theoretical discussions, 20 min for mindfulness exercises, and 20 min for learning-related activities. As we progressed to Week 3 and Week 4, the time allocation shifted to 10 min for theoretical discussion, 15 min for mindfulness, and 35 min for learning-related activities. Mindful learning activities were designed with specific goals. The first involved exploring students’ values to enhance intrinsic motivation. The second introduced the 4D cycle in appreciative inquiry, aiming to cultivate self-management. Week 3 focused on personal and group narratives, developing learning engagement, fostering goals, and promoting self-directed learning. In Week 4, the learning activities centered on exploring generative moments, revealing character strengths, and self-evaluating growth to cultivate emotion regulation and track learner identity progress. In addition, the participants were required to practice mindfulness for five to ten minutes every day during the intervention period, using the audio-recorded instructions of mindfulness-based practice (e.g., mindful breathing, mindful eating, body scan, and compassion meditation) that were sent by the researchers via a social media application on smart phones. The instructions were recorded by one of the authors, who had 10 years’ experience in teaching and practicing mindfulness. The participants were asked to report their daily practice in online chat rooms, for them to receive peer support from other participants and encouragement from the researchers. Three participants (20%) practiced at least three times a week, and the majority of the participants practiced twice or once a week. 

The participants in the active control group were invited to a relaxation-training program to eliminate the influence of the placebo effect in the intervention [100]. This program was designed similarly to the MLC, and comprised four one-hour workshops, one per week (see Table 7 for the detailed activities and aims). The participants were additionally required to practice relaxation for five to ten minutes every day during the intervention period. The assignment procedure was the same as that in the experimental group. Five participants (33.3%) practiced at least three times a week.

The participants in the blank control group did not receive any form of intervention. All the participants completed online questionnaires on mindful agency, SWB, and PWB twice at approximately the same periods of time when the participants in the experimental and active control groups completed the measurements. 

### 3.4. Data Analysis

Two-way repeated-measures ANOVA and a paired-samples *t*-test were used to analyze the data. The between-subject variables included group (experimental vs. active control vs. blank control groups) and within-subject variables included time (pre-test vs. post-test). Since age, gender, and meditation experience were found to differ significantly in some sub-dimensions of SWB and PWB (i.e., autonomy and positive affect), they were added into the models as covariables in the data analysis. The dependent variables were the scores on mindful learning, SWB, and PWB. The significance level was set at *p* < 0.05 to minimize Type 1 errors.

### 3.5. Results

There were no significant group differences in the baseline case (mindful learning: *F* (2, 45) = 0.879, *p* = 0.422; SWB: *F* (2, 45) = 0.855, *p* = 0.423; PWB: *F* (2, 45) = 2.189, *p* = 0.124). In the experimental group, we conducted Mann–Whitney U tests and found no significant differences in the changes of mindful learning total score (*p* = 0.055), Psychological Well-Being (PWB) total score (*p* = 0.594), and Subjective Well-Being (SWB) total score (*p* = 0.371) between participants with and without prior meditation experience. The means, standard deviations, and repeated-measures ANOVA results for mindful learning, positive affect, and PWB at T1 and T2 in the three groups are presented in Table 8. Generally, H3, that the participants in the experimental group significantly increased SWB and PWB compared with those in the active and blank control groups, was supported.

#### 3.5.1. The Development of Mindful Learning 

The results of the repeated-measures ANOVA are presented in Table 8. There was a significant interaction effect between group and time [*F* (2, 42) = 5.049, *p* = 0.011, *η*^2^ = 0.194] on mindful learning. Further simple-effect analysis found a significant improvement (*p* = 0.010) in mindful learning in the experimental group after MLC, while there was no significant change in mindful learning in both control groups from T1 to T2 (see Figure 2). This result indicated that MLC successfully enhanced mindful learning. 

#### 3.5.2. The Development of SWB

A detailed repeated-measures ANOVA was conducted on the dimensions of SWB (i.e., life satisfaction, positive affect, and negative affect). The results showed a significant main effect of time [*F* (1, 42) = 3.072, *p* = 0.057, *η*^2^ = 0.128] on positive affect, while there was no significant interaction or main effect on the other dimensions (*p* > 0.05). The paired-samples *t*-test showed that positive affect significantly improved in the experimental group after coaching (*t* = 4.179, *p* < 0.001), compared with its non-significant decrease in the active-(*t* = 0.507, *p* = 0.620) and blank control groups (*t* = 0.236, *p* = 0.816). The changes in all the SWB dimensions in the different groups from T1 to T2 are displayed in Figure 3.

#### 3.5.3. The Development of PWB

The results of the repeated-measures ANOVA of PWB indicated a significant interaction effect between group and time [*F* (2, 42) = 6.566, *p* = 0.003, *η*^2^ = 0.238]. Further simple-effect analysis showed that the participants’ PWB significantly decreased in both active- (*p* = 0.008) and blank control groups (*p* = 0.009) at T2. For the experimental group, although there was no significant difference (*p* = 0.190) in PWB from T1 to T2, clearly the participants’ PWB tended to increase after coaching (see Figure 4).

#### 3.5.4. The Impact of Mindful Learning on SWB and PWB

To further understand the effect of mindful learning on SWB and PWB, we calculated Spearman’s correlation between the study’s variables in the intervention group (Table 9). Significant correlations were observed between the change of total scores of mindful learning and PWB (*r* = 0.74, *p* = 0.002). A detailed investigation revealed that the development of openness to experience was related to improvements in the positive affect (*r* = 0.63, *p* = 0.012) in SWB and PWB (*r* = 0.69, *p* = 0.005). Also, the development of learning engagement was related to improvements in the life satisfaction (*r* = 0.67, *p* = 0.010) in SWB.

To further verify the impact of mindful learning on PWB, we programmed three groups into dummy variables (experimental group = 2, active control group= 1, blank control group = 0) and used Model 4 of PROCESS [101] to conduct an analysis, with change in PWB as the dependent variable and change in mindful learning as the mediator. The results showed that the intervention had a significant influence on mindful learning (*B* = 3.10, *t* = 3.178, *p* = 0.003) and PWB (*B* = 2.653, *t* = 2.66, *p* = 0.011). When change in mindful learning was included for mediation analysis, it had a significant effect on PWB (*B* = 0.56, *t* = −2.72, *p* = 0.009). Change in mindful learning fully mediated the effect of intervention on PWB (see Figure 5). Therefore, the hypothesis (H4), that change in mindful learning led to the development of PWB and SWB, was partially supported after the coaching intervention. 

## 4. Discussion

The results of Study 1 showed that the postgraduate students’ mindful learning, SWB, and PWB were positively associated and that mindful learning could positively predict SWB and PWB. Specifically, mindful learning was moderately associated with life satisfaction and positive affect and negatively associated with negative affect. The findings of Study 2 showed that after MLC, the coaching intervention designed to enhance mindful learning, the participants in the experimental group significantly improved mindful learning compared with those in both the control groups. The participants in the experimental group experienced increases in positive affect and PWB, while the students in the other two groups experienced decreases in PWB. In addition, changes in mindful learning fully mediated the effect of MLC on the students’ PWB. Overall, these findings indicate that mindful learning can influence students’ well-being, with a stronger impact on PWB than on SWB. 

First, the results of Study 1 demonstrate strong theoretical connections between mindful learning and well-being and perhaps a stronger link between mindful learning and PWB. Mindful learning is related to SWB because students’ higher autonomous motivation in learning is accompanied by stronger positive emotions [63,102]. An active learning attitude is conducive to pleasant, positive learning emotions [103,104]. The regression analysis showed that, of the mindful learning dimensions, learning methods and emotion regulation impacted SWB. Students with a high level of mindful learning can adopt appropriate learning strategies and methods according to circumstances, regulate and balance positive and negative emotions during learning, and identify learning opportunities instead of being constrained by external environments; thus, they are more active in creating satisfying learning environments [64,81]. Therefore, a higher-level mindful learning is associated with higher levels of satisfaction and positive affect. Of all the dimensions of SWB, mindful learning showed the weakest relationship with negative affect. This finding resonates with previous studies on SWB that have found that positive and negative affects are two related but distinct aspects of emotions [29,30,97,105,106]. A higher level of mindful learning is strongly related to increased student satisfaction and positive emotions; however, it may not be very strong in terms of reducing negative affect. Indeed, experiencing negative emotions such as disturbance, anxiety, and stress during learning is common, and knowing how to manage and even use negative affect is an important aspect of mindful learning (i.e., emotion regulation) [81].

Mindful learning showed a moderate, positive relationship with PWB. Such a robust link is easy to understand, given that the notion of mindful learning is conceptually associated with all the dimensions of PWB. For example, self-acceptance is a central feature of psychological health and a characteristic of self-actualization in PWB [34] and is addressed in the mindfulness theory that underlines mindful learning [10]. Autonomy in PWB means that individuals make their own decisions based on their own judgments and values, have an internal locus of evaluation, and do not look to others for approval [34,35,68,107,108]. This dimension is similar to autonomy as defined in SDT; that is, that individuals authentically endorse their actions and make choices that reflect their true interests and values [39]. Since SDT is the theoretical foundation of mindful learning, clearly mindful learners address the freedom to act in accordance with their interests and values in making authentic learning decisions [81]; thus, they rate highly in autonomy and obtain higher levels of PWB [68]. Environmental mastery, another PWB dimension that is closely related to mindful learning, refers to an individual’s ability to choose or create environments suitable to their psychological conditions [34,35]. Mindful learners are better at knowing how to take advantage of environmental opportunities, create and optimize learning environments, and actively participate in and shape the environment that is important for their learning [64,81]. Overall, mindful learning is associated with an increased level of PWB.

Interestingly, the strongest predictor of PWB and SWB in the mindful learning dimensions was learning methods. Knowing when and how to adopt appropriate learning strategies is perhaps one of the most critical elements in learning how to learn [109]. The ability to utilize different methods accordingly and flexibly in various learning activities and within various learning contexts is a comprehensive, higher-order capacity that reflects learners’ self-regulatory and metacognitive levels [81]. Compared to the other dimensions (openness to experience or learning engagement), which may be the more attitudinal aspects of mindful learning, learning methods may be more goal-focused and action-oriented. In other words, individuals are affectively and cognitively ready to take actions in learning by thinking about the best way to conduct learning activities. This forward-looking, goal-focused disposition shares several core characteristics with proactive coping [110,111] in that they both highlight in advance actions, goal management, self-regulation, and perceived control and self-efficacy, which are effective methods for enhancing well-being. 

Second, the results of Study 2 showed that MLC could significantly facilitate postgraduate students’ development of mindful learning, consistent with previous findings [64,73,74,81,112,113,114]. Numerous coaching activities directly addressed students’ positive resources and thus enabled increases in mindful learning and well-being. For instance, coaching tasks that explored learners’ values, strengths, and inner motivations enabled students to take a positive view of themselves and enhance self-esteem and self-efficacy; these variables were found to be significantly related to satisfaction, learning engagement, sense of purpose, positive emotions, and stress coping [87,115,116]. Cultivating attitudes of mindfulness enhanced students’ self-awareness and self-acceptance, for them to embrace moment-to-moment sensational experiences, thoughts, purposes, and emotions openly and non-judgmentally [64,81,88]. Narrative coaching facilitated students’ development of positive personal stories and self-congruence and strengthened self-directed learning [64,81,117]. Overall, MLC was effective in increasing the students’ mindful learning that led to a higher level of self-efficacy, self-awareness, self-determination, and self-regulation through which they experienced higher SWB and PWB [81,107,116,118].

Third, the results of Study 2 showed that changes in mindful learning led to the development of PWB. This effect can be explained by SDT and mindfulness theory, as proposed in the theoretical framework. The MLC program successfully cultivated the students’ mindful learning, and these students could adjust their learning motivation more autonomously, were more active and independent in the process of realizing their learning goals, and achieved self-realization [108,118]. Studies have found that higher autonomy leads to higher autonomous motivation and PWB [54,119]. In addition, mindfulness in learning helps improve individual performance, promotes personal development, improves physical and mental health, and relieves stress [49], thus leading to higher PWB.

Openness to experience is particularly important in the development of SWB and PWB. When mindful learners show openness to various learning experiences and acknowledge them as a natural part of learning, they are more receptive to setbacks, pay more attention to the gains previously overlooked, and therefore may experience more positive affects in SWB [81]. Moreover, mindful learners can reduce cognitive rigidity and increase mental flexibility through openness to experience [48], view themselves as growing and developing [35], and ascend in competence in managing the demands of life [120,121], thus increasing personal growth and environmental mastery in PWB. 

Nonetheless, we must point out that the enhancement of mindful learning influenced the students’ SWB and PWB to a certain extent. Study 2 was conducted close to the final week of the academic term. It is plausible that students may have experienced a lower general well-being due to the heavy workload of assignments and final exams [122]. Therefore, it appears that the decrease in well-being was more pronounced in PWB, which may explain why PWB decreased in both control groups. In contrast, there were distinct increases in positive affect and PWB for the students in the experimental group. This result demonstrated that the development of mindful learning through MLC could at least prevent a decrease in well-being and increase students’ well-being to some extent in highly stressful circumstances. Therefore, mindful learning may be considered a protective factor in students’ general well-being. However, it is crucial to acknowledge the relatively small sample size despite meeting the requirements of the power analysis and to be aware of the possibility of false negatives, where effects might exist at the population level but may not reach statistical significance in the results.

### Limitations and Future Directions

The current study pioneers an investigation into the influence of mindful learning on postgraduate students’ well-being and may advance the literature on linking learning theories with positive psychology. However, we must address several limitations in the study: the small sample size and homogeneity; the use of only self-reported measures; low Cronbach’s alphas of the subscales of PWB; using a relatively short-term version of the MLC. The low rate of participation in daily mindfulness practice among the participants is notable. This could be due to several reasons, such as difficulty in incorporating the practice into their daily routines, a lack of motivation, or even adverse experiences with the practice. This aspect of our study warrants further attention, as it might indicate potential challenges in promoting sustainable mindfulness practice and could provide useful insights for future research. Future studies should invite more participants, use more properly validated psychometric tools, incorporate more objective and reliable measures of improvement (e.g., academic performance), and adopt methodological strategies (e.g., using a coaching manual and recording sessions for reflection) that enhance reliable and consistent intervention delivery during coaching trials. Although the STD and mindfulness theory formed the theoretical basis, the mechanism of the influence that mindful learning exerts on well-being requires further examination. Future research could delve into exploring and quantifying the specific mechanisms related to SDT, such as measuring the variables of autonomy, competence, and relatedness. Furthermore, the selection of the control group could be further optimized in future research. While we utilized an active control group, it is acknowledged that a more refined comparison could involve a group receiving general mindful training. This would provide a more comprehensive understanding of the unique effects of our intervention compared to a broader mindfulness context. Finally, future studies can employ mixed methods in data collection and analysis to enhance scientific rigor.

## 5. Conclusions

The current study used a cross-sectional design and a randomized controlled trial of intervention to investigate the association between postgraduate students’ mindful learning and both their subjective and psychological well-being as well as the role of mindful learning in students’ well-being. The results of Study 1 demonstrated robust theoretical and empirical correlations of mindful learning, life satisfaction, positive affect, and PWB. The strongest predictor of both PWB and SWB among mindful learning dimensions was learning methods, which underscores the importance of being mindful during the learning process and adopting appropriate learning strategies. In Study 2, the Mindful Learning Coaching (MLC) intervention was found to significantly enhance students’ mindful learning. The participants who received coaching increased both SWB and PWB significantly, while the participants in the two control groups experienced decreases in these measures. Our results demonstrate that the development of mindful learning through MLC could, to some extent, counteract the decrease in well-being in highly stressful circumstances, indicating that mindful learning is a protective factor for students’ general well-being. In summary, our study advances the existing knowledge of mindful learning and contributes to empirical exploration of its impact on well-being. Nevertheless, it is essential to note that the relationship between mindful learning and well-being can be influenced by various factors, and more empirical efforts are called for to gain an in-depth understanding of the mechanisms. 

## Figures and Tables

**Figure 1 behavsci-13-01009-f001:**
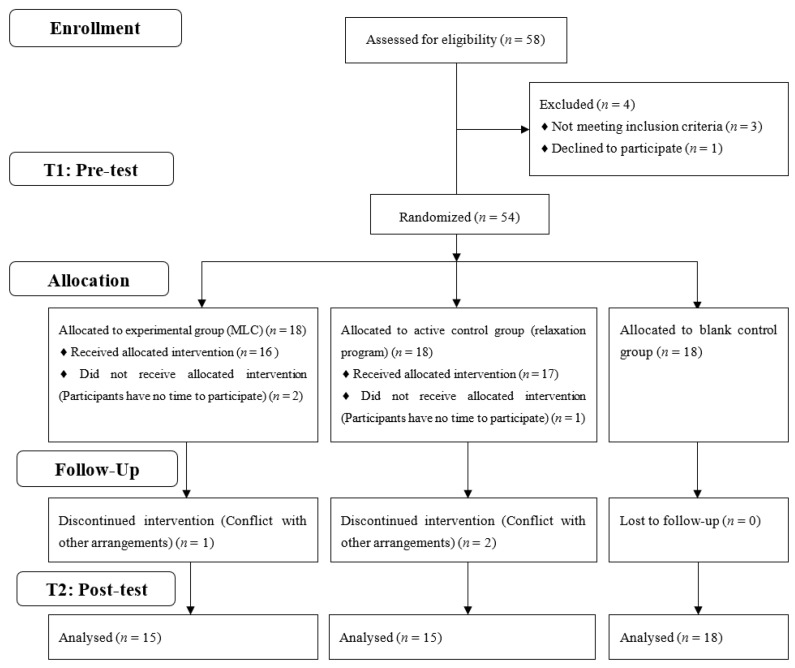
Participants’ flow and the procedure for Study 2.

**Figure 2 behavsci-13-01009-f002:**
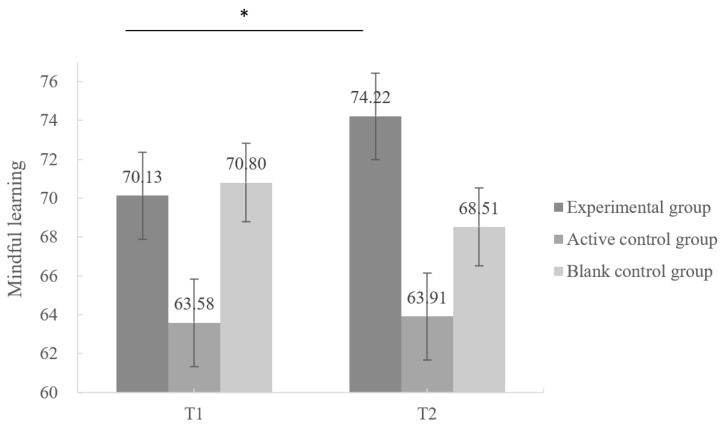
Changes in postgraduates’ mindful learning in different groups from baseline to post-test. Note. * *p* < 0.05. The error bars represent standard errors (SE).

**Figure 3 behavsci-13-01009-f003:**
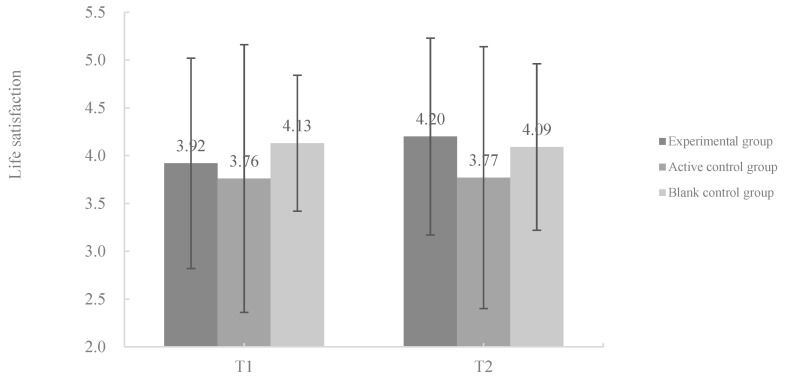
Changes in postgraduates’ SWB in different groups from baseline to post-test. Note. ** *p* < 0.01. The error bars represent standard errors (SE).

**Figure 4 behavsci-13-01009-f004:**
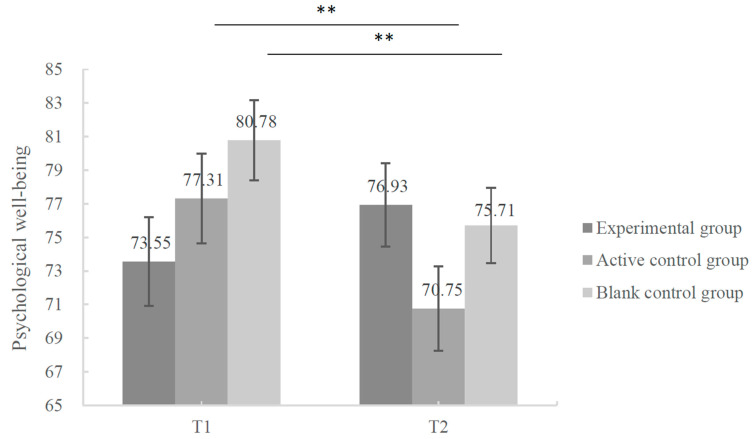
Changes in all postgraduates’ SWB dimensions in different groups from baseline to post-test. Note. ** *p* < 0.01. The error bars represent standard errors (SE).

**Figure 5 behavsci-13-01009-f005:**
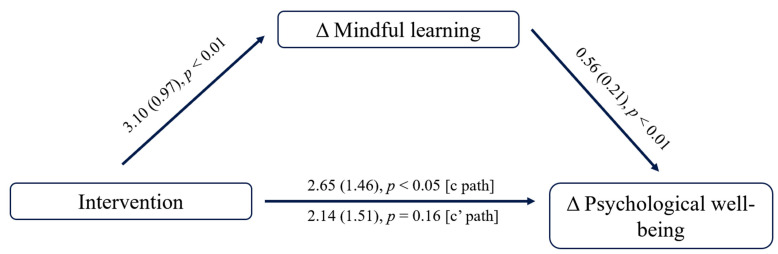
Changes in postgraduates’ PWB in different groups from baseline to post-test. Notes. Δ indicates pre-post changes (post-test minus pre-test).

**Table 1 behavsci-13-01009-t001:** Description of respondents by gender and meditation practice experience.

	Cronbach’s Alpha	Male (*n* = 80)		Female (*n* = 156)		*t*	Meditation Experience (*n* = 104)		No Meditation Experience (*n* = 132)		*t*
		*Mean*	*SD*	*Mean*	*SD*		*Mean*	*SD*	*Mean*	*SD*	
1.Mindful learning	0.85	70.63	11.78	69.71	9.12	0.61	70.24	10.05	69.85	10.15	0.30
1.1 Learning methods	0.72	14.48	3.06	14.07	2.26	1.04	14.16	2.53	14.24	2.60	−0.23
1.2 Emotion regulation	0.66	8.10	2.28	7.81	2.15	0.95	7.94	2.06	7.89	2.31	0.19
1.3 Awareness of planning	0.66	13.19	3.01	13.02	2.33	0.44	13.16	2.47	13.01	2.67	0.46
1.4 Openness to experience	0.68	21.64	3.99	21.79	3.52	−0.30	21.77	3.91	21.71	3.49	0.12
1.5 Learning engagement	0.60	13.23	2.99	13.02	2.38	0.53	13.20	2.50	13.00	2.68	0.59
2.Subjective well-being											
2.1 Life satisfaction	0.84	4.21	1.38	4.05	1.15	0.87	4.18	1.29	4.05	1.18	0.76
2.2 Positive affect	0.88	4.64	1.34	4.37	1.28	1.53	4.74	1.21	4.24	1.34	2.93 **
2.3 Negative affect	0.86	3.11	1.34	3.21	1.18	−0.57	3.24	1.28	3.12	1.20	0.74
3.Psychological well-being	0.82	75.65	11.47	74.12	9.93	1.06	74.97	10.45	74.38	10.52	0.43
3.1 Autonomy	0.36	12.23	2.32	11.18	2.09	3.50 **	11.71	2.20	11.39	2.24	1.09
3.2 Environmental mastery	0.55	12.99	2.63	12.64	2.22	1.07	12.64	2.39	12.85	2.35	−0.66
3.3 Self-acceptance	0.49	11.68	2.59	11.13	2.37	1.61	11.44	2.42	11.22	2.49	0.69
3.4 Personal growth	0.49	14.00	2.68	13.99	2.24	0.02	14.02	2.24	13.98	2.51	0.13
3.5 Positive relations	0.55	11.70	3.05	12.18	2.67	−1.24	12.26	2.85	11.83	2.77	1.18
3.6 Purpose in life	0.52	13.06	2.92	12.99	2.72	0.18	12.89	2.76	13.11	2.81	−0.60

Notes. ** *p* < 0.01.

**Table 2 behavsci-13-01009-t002:** Descriptive and Pearson correlations of the variables in Study 1 (*n* = 236).

	1	1.1	1.2	1.3	1.4	1.5	2.1	2.2	2.3	3
**1.Mindful learning**										
1.1 Learning methods	0.72 **									
1.2 Emotion regulation	0.60 **	0.27 **								
1.3 Awareness of planning	0.80 **	0.47 **	0.55 **							
1.4 Openness to experience	0.81 **	0.50 **	0.29 **	0.49 **						
1.5 Learning engagement	0.73 **	0.40 **	0.24 **	0.49 **	0.52 **					
**2.Subjective well-being**										
2.1 Life satisfaction	0.40 **	0.25 **	0.39 **	0.34 **	0.29 **	0.24 **				
2.2 Positive affect	0.42 **	0.35 **	0.26 **	0.33 **	0.31 **	0.30 **	0.54 **			
2.3 Negative affect	−0.28 **	−0.29 **	−0.09	−0.20 **	−0.23 **	−0.19 **	−0.27 **	−0.38 **		
**3.Psychological well-being**	0.49 **	0.47 **	0.23 **	0.38 **	0.39 **	0.31 **	0.31 **	0.40 **	−0.41 **	
** *Mean* **	70.02	14.21	7.91	13.08	21.74	13.09	4.11	4.46	74.64	11.53
** *SD* **	10.09	2.56	2.20	2.58	3.68	2.60	1.23	1.30	10.47	2.22

Notes. *n* = 236, ***p <* 0.01 (two-tailed).

**Table 3 behavsci-13-01009-t003:** (**a**) Linear and multiple regression analyses of mindful learning and its dimensions on PWB. (**b**) Linear and multiple regression analyses of mindful learning and its dimensions on SWB.

**(a)**									
**Outcome Variable**	**Predictor Variable**	** *B* **	** *SE* **	** *β* **	** *F* **	** *R²* **	** *f²* **	** *Adjusted R²* **	** *t* **
PWB		42.742	9.283		18.217 ***	0.240	0.32	0.227	4.604 ***
	Mindful learning	0.502	0.060	0.484					8.423 ***
	Gender	−1.166	1.291	−0.053					−0.903
	Meditation experience	−0.564	1.224	−0.027					−0.461
	Age	−0.020	0.308	−0.004					−0.064
PWB		41.818	9.221		10.570 ***	0.271	0.37	0.246	4.535 ***
	Learning methods	1.278	0.283	0.313					4.520 ***
	Emotion regulation	0.101	0.325	0.021					0.309
	Awareness of planning	0.553	0.327	0.136					1.689
	Openness to experience	0.382	0.210	0.134					1.823
	Learning engagement	0.163	0.283	0.040					0.575
	Gender	−1.042	1.282	−0.047					−0.813
	Meditation experience	−0.701	1.210	−0.033					−0.580
	Age	−0.041	0.306	−0.008					−0.135
**(b)**									
**Outcome Variable**	**Predictor Variable**	** *B* **	** *SE* **	** *β* **	** *F* **	** *R²* **	** *f²* **	** *Adjusted R²* **	** *t* **
Life satisfaction		−0.239	1.136		12.03 ***	0.17	0.21	0.16	−0.210
	Mindful learning	0.048	0.007	0.396					6.605 ***
	Gender	−0.104	0.158	−0.040					−0.658
	Meditation experience	−0.109	0.150	−0.044					−0.726
	Age	0.055	0.038	0.088					1.465
Positive affect		0.348	1.161		17.41 ***	0.23	0.30	0.22	0.299
	Mindful learning	0.053	0.007	0.410					7.094 ***
	Gender	−0.272	0.161	−0.099					−1.686
	Meditation experience	−0.498	0.153	−0.190					−3.257 **
	Age	0.069	0.039	0.104					1.790
Negative affect		5.630	1.205		4.93 **	0.08	0.09	0.06	4.672 ***
	Mindful learning	−0.034	0.008	−0.275					−4.348 ***
	Gender	0.047	0.168	0.018					0.283
	Meditation experience	−0.126	0.159	−0.051					−0.795
	Age	0.001	0.040	0.002					0.024
Life satisfaction		0.019	1.130		7.31 ***	0.21	0.27	0.18	0.017
	Learning methods	0.027	0.035	0.057					0.791
	Emotion regulation	0.156	0.040	0.280					3.921 ***
	Awareness of planning	0.032	0.040	0.067					0.792
	Openness to experience	0.043	0.026	0.128					1.664
	Learning engagement	0.020	0.035	0.043					0.589
	Gender	−0.092	0.157	−0.035					−0.584
	Meditation experience	−0.109	0.148	−0.044					−0.733
	Age	0.049	0.037	0.079					1.319
Positive affect		0.318	1.173		8.87 ***	0.24	0.32	0.21	0.271
	Learning methods	0.099	0.036	0.195					2.760 **
	Emotion regulation	0.063	0.041	0.107					1.532
	Awareness of planning	0.034	0.042	0.068					0.822
	Openness to experience	0.033	0.027	0.092					1.224
	Learning engagement	0.053	0.036	0.107					1.484
	Gender	−0.252	0.163	−0.092					−1.547
	Meditation experience	−0.503	0.154	−0.192					−3.267 **
	Age	0.066	0.039	0.100					1.710
Negative affect		5.801	1.208		3.15 **	0.10	0.11	0.07	4.800 ***
	Learning methods	−0.105	0.037	−0.217					−2.829 **
	Emotion regulation	0.018	0.043	0.032					0.424
	Awareness of planning	−0.026	0.043	−0.053					−0.597
	Openness to experience	−0.026	0.027	−0.078					−0.958
	Learning engagement	−0.021	0.037	−0.044					−0.561
	Gender	0.038	0.168	0.015					0.226
	Meditation experience	−0.115	0.159	−0.046					−0.724
	Age	0.001	0.040	0.001					0.019

Notes. *N* = 236, *** *p < 0*.001, ** *p* < 0.01 (two-tailed).

**Table 4 behavsci-13-01009-t004:** The number and proportion of each gender in the three groups.

Gender	Experimental Group		Active Control Group		Blank Control Group		*F*	*p*
	*n* = 15	Proportion	*n* = 15	Proportion	*n* = 18	Proportion		
Male	6	40.00%	7	46.67%	6	33.33%	0.290	0.750
Female	9	60.00%	8	53.33%	12	66.67%		

**Table 5 behavsci-13-01009-t005:** Cronbach’s α for Mindful learning, SWB, and PWB in Study 2.

	Cronbach’s α in Pre-Test	Cronbach’s α in Post-Test
Mindful learning	0.856	0.880
Life satisfaction	0.784	0.842
Positive affect	0.896	0.903
Negative affect	0.799	0.868
Psychological well-being	0.668	0.648

**Table 6 behavsci-13-01009-t006:** Mindful Learning Coaching (MLC) in the experimental group.

Session	Main Activities	Aims
Session 1 (1 h)	·Getting to know each other and building group spirit ·Introducing the concept of mindful learning ·Explaining mindfulness and experiencing being in the present ·Exploring students’ values, recourses, reasons, and motivation in learning	·Foster a sense of agency in learning ·Enhance intrinsic motivation in learning
Session 2 (1 h)	·Preparing the attitudes of mindfulness practice ·Experiencing mindful breathing and mindful eating ·Introducing 4D cycle in appreciative inquiry and sharing in pairs	·Develop self-awareness ·Deepen reflexivity ·Cultivate self-management and self-discipline
Session 3 (1 h)	·Experiencing mindful body scanning ·Discussing narrative psychology ·Personal narrative: Writing a letter for yourself in 10 years ·Narrative in pairs: Reconstruction of stories ·Group narrative: A journey to the West	·Develop learning engagement ·Cultivate a sense of goal, aspiration, and hope ·Develop self-directed and collaborative learning ability
Session 4 (1 h)	·Experiencing loving-kindness meditation on learning ·Exploring generative moments in learning ·Revealing character strengths ·Reviewing, sharing, and reflecting on the coaching ·Self-evaluation of growth as a learner	·Foster self-knowledge and self-awareness ·Cultivate the ability of emotion regulation in learning ·Track the progress of learners’ identities
Assignment	Daily mindfulness practice (5–10 min)	·Foster reflexivity, cultivate openness to experience, and non-judgmental attitudes ·Develop learning attention in the present moment

**Table 7 behavsci-13-01009-t007:** Relaxation training program in the active control group.

Session	Main Activities	Aims
Session 1 (1 h)	·Getting to know each other and building group spirit ·Introducing relaxation training and practicing abdominal breathing ·Game: Grabbing Fingers; Nerve Conduction	·Form a harmonious group atmosphere ·Help participants relax
Session 2 (1 h)	·Reviewing abdominal breathing; practicing progressive muscle relaxation training ·Game: Team up for relay drawing ·Watching short cartoon	·Develop a sense of cooperation ·Ease participants’ moods
Session 3 (1 h)	·Practicing count-breathing relaxation and eye relaxation ·Game: Numbering off without communication; leaf drawing	·Establish trust with peers ·Relieve anxiety
Session 4 (1 h)	·Practicing pure release relaxation ·Game: Team up for building model with A4 paper ·Reviewing, sharing, and ending	·Promote communication with others ·Track the progress of participants’ relaxation experiences
Assignment	Daily relaxation practice (5–10 min)	·Develop a sense of relaxation and relieve anxiety

**Table 8 behavsci-13-01009-t008:** Means and standard deviations of variables and results of covariance analysis at T1 and T2 in Study 2.

	Experimental Group (*n* = 15)		Active Control Group (*n* = 15)		Blank Control Group (*n* = 18)		Time × Group		
	T1	T2	T1	*T2*	T1	T2	*F*	*p*	*η* ^2^
Mindful learning	68.93 ± 9.77	72.87 ± 9.65	65.53 ± 11.40	66.00 ± 11.72	70.17 ± 9.51	67.89 ± 9.23	5.049	0.011	0.194
Life satisfaction	3.92 ± 1.10	4.20 ± 1.03	3.76 ± 1.40	3.77 ± 1.37	4.13 ± 0.71	4.09 ± 0.87	0.905	0.412	0.041
Positive affect	3.86 ± 1.27	4.57 ± 1.21	3.89 ± 1.33	3.97 ± 1.37	4.03 ± 1.26	4.10 ± 1.28	3.072	0.057	0.128
Negative affect	3.74 ± 0.89	3.58 ± 0.92	3.33 ± 1.11	3.49 ± 1.28	3.00 ± 0.88	3.07 ± 1.03	0.702	0.501	0.032
PWB	73.00 ± 8.72	75.80 ± 7.68	78.20 ± 13.69	72.40 ± 12.72	80.50 ± 8.30	75.27 ± 10.10	6.566	0.003	0.238

Notes. Covariance analysis was performed using age, gender, and meditation experience as covariables.

**Table 9 behavsci-13-01009-t009:** Spearman’s correlations amongst changes in study variables.

	1	1.1	1.2	1.3	1.4	1.5	2.1	2.2	2.3
1. ΔMindful learning									
1.1 Δ Learning methods	0.61 *								
1.2 Δ Emotion regulation	0.61 *	0.19							
1.3 Δ Awareness of planning	0.57 *	0.15	0.36						
1.4 Δ Openness to experience	0.70 **	0.63 *	0.26	−0.03					
1.5 Δ Learning engagement	0.46	0.01	0.03	0.36	0.11				
2. ΔSubjective well-being									
2.1 Δ Life satisfaction	−0.04	0.06	−0.42	−0.10	−0.11	0.67 **			
2.2 Δ Positive affect	0.55 *	0.46	0.23	−0.15	0.63 *	0.14	0.09		
2.3 Δ Negative affect	−0.08	−0.43	−0.09	0.40	−0.21	0.10	−0.27	−0.46	
3.ΔPsychological well-being	0.72 **	0.57 *	0.35	0.17	0.69 **	0.32	0.20	0.47	−0.47

Notes. * *p* < 0.05 and ** *p* < 0.01; *n* = 15; Δ indicates pre-post changes (post-test minus pre-test).

## Data Availability

The datasets generated during and/or analyzed during the current study are available from the corresponding author on reasonable request.

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
