# Peer review of "The Impact of Mindful Learning on Subjective and Psychological Well-Being in Postgraduate Students"

_behavsci, 2023, doi:10.3390/bs13121009_

Round 1
Reviewer 1 Report
Comments and Suggestions for Authors
The manuscript presents a comprehensive study on the impact of mindful learning on subjective and psychological well-being among postgraduate students. The theoretical grounding and methodology are generally robust, and the findings contribute to the literature on mindfulness and well-being in educational settings. I believe, however, that addressing these points will further strengthen the manuscript.
- From the MLC plan, it seems that only a small part of the activities is directly related to learning. It is thus unclear whether it was mindful learning or general mindful training that contributed to PWB. A better control than the Activity control group used in this study may be a group that receive general mindful training.
- In Figure 5, why B is negative from Intervention to both Delta_Mindful learning and Delta_PWB? Does this mean intervention has negative effects? Why is this?
- It is not very clear how SDT is related to the observations of this study, given that there is no direct measurement of relatedness and competence in the present study.
- Please add legend for errorbars to all the figures.
English is fine.
Author Response
Reviewer #1:
- From the MLC plan, it seems that only a small part of the activities is directly related to learning. It is thus unclear whether it was mindful learning or general mindful training that contributed to PWB.
Response:
Thank you for your insightful comments. We have provided additional details to elucidate the learning-related components in the MLC plan on page 10.
“During the initial two weeks, our sessions followed a 20-20-20 structure: 20 minutes for theoretical discussions, 20 minutes for mindfulness exercises, and 20 minutes for learning-related activities. As we progressed to Week 3 and Week 4, the time allocation shifted to 10 minutes for theoretical discussion, 15 minutes for mindfulness, and 35 minutes for learning-related activities. Mindful learning activities were designed with specific goals. The first involved exploring students' values to enhance intrinsic motivation. The second introduced the 4D cycle in appreciative inquiry, aiming to cultivate self-management. Week 3 focused on personal and group narratives, developing learning engagement, fostering goals, and promoting self-directed learning. In Week 4, the learning activities centered on exploring generative moments, revealing character strengths, and self-evaluating growth to cultivate emotion regulation and track learner identity progress.”
- A better control than the Activity control group used in this study may be a group that receive general mindful training.
Response:
We appreciate your insightful suggestion regarding the control group in our study. Recognizing the need for a more robust control, we acknowledge the limitation in our current design.
In the “Limitations and future directions” section of our manuscript, we have included a statement addressing this concern on page 18.
“Furthermore, the selection of the control group could be further optimized in future research. While we utilized an active control group, it is acknowledged that a more refined comparison could involve a group receiving general mindful training. This would provide a more comprehensive understanding of the unique effects of our intervention compared to a broader mindfulness context.”
- In Figure 5, why B is negative from Intervention to both Delta_Mindful learning and Delta_PWB? Does this mean intervention has negative effects? Why is this?
Response:
Thank you for your observation regarding Figure 5. The negative values observed in B from Intervention to both Delta_Mindful learning and Delta_PWB were indeed a result of our initial coding strategy.
Originally, we coded the experimental group as 1, the active control group as 2, and the blank control group as 3 when creating dummy code variables, leading to a less intuitive interpretation. We sincerely thank you for bringing this to our attention. In response, we have revised the group coding to enhance clarity. The new coding is as follows: experimental group = 2, active control group = 1, blank control group = 0. We have included an explicit explanation of this coding adjustment in the manuscript to ensure transparency. Consequently, the revised results now indicate that the intervention had a positive and significant influence on mindful learning (B = 3.10, t = 3.178, p = 0.003) and psychological well-being (B = 2.653, t = 2.66, p = 0.011). Please see page 15 for the revision.
- It is not very clear how SDT is related to the observations of this study, given that there is no direct measurement of relatedness and competence in the present study.
Response:
Thanks for your comment. In our study, SDT serves as a foundational element in the theoretical framework of mindful learning along with mindfulness theory. As discussed on page 3, SDT recognizes individuals' potential for self-realization and autonomy, with three basic needs (autonomy, relatedness, and competence) being key to optimal functioning and well-being. Mindful learning is particularly related to the autonomy and competence needs, and the importance of relatedness is stated in both learning and well-being that aligns with mindful learning principles. SDT provides a comprehensive understanding how mindful learning influences well-being through autonomous motivation and needs satisfaction. However, the components of SDT was not suitable for direct measurement in the current study since we have integrated the theory with mindfulness theory into the conceptualization of mindful learning. In order to address your concern, we have added an acknowledgement in the “Limitations and Future Directions” section on page 18:
“Although the STD and mindfulness theory formed the theoretical basis, the mechanism of the influence that mindful learning exerts on well-being requires further examination. Future research could delve into exploring and quantifying the specific mechanisms related to SDT, such as measuring the variables of autonomy, competence, and relatedness.”
- Please add legend for errorbars to all the figures.
Response:
Thank you for your suggestion regarding the addition of legends for error bars in all figures. We have addressed this issue by including the following note: “The error bars represent standard errors (SE).”
Reviewer 2 Report
Comments and Suggestions for Authors
The study encompasses both survey methods and an experimental design, providing a comprehensive perspective.
The positive correlation between mindful learning, subjective well-being (SWB), and psychological well-being (PWB) is a significant contribution. However, there is a debate in the scientific literature that shows SWB and PCB as the same concept, so it is not so much understood to use two different constructs that are so similar to each other as several measures of the same concept that is related to mindful learning.
The use of regression analysis and random assignment in the experimental group reinforces methodological robustness. The key findings of your paper are that Mindful learning coaching showed significant improvements in students' mindful practice and that the Control groups showed decreases in SWB and PWB, while the experimental group tended to experience increases. But it could be helpful to explore more deeply the reasons behind the decrease in SWB and PWB in the control groups, probably based on the low practice on daily life.
However, there are two limitations: The sample size in the experimental group (54 students) might be considered relatively small. It would be helpful to discuss the implications of this sample size in the conclusions, not only identify the limitation.
Author Response
Reviewer #2:
The sample size in the experimental group (54 students) might be considered relatively small. It would be helpful to discuss the implications of this sample size in the conclusions, not only identify the limitation.
Response:
Thank you for your suggestion. Despite meeting the requirements of our power analysis (n = 42) with 54 participants, it is essential to recognize the relative smallness of the sample. We have added the implications of the sample size in the Discussion section on page 18.
“However, it is crucial to acknowledge the relatively small sample size despite meeting the requirements of the power analysis and to be aware of the possibility of false negatives, where effects might exist at the population level but may not reach statistical significance in the results.”
Reviewer 3 Report
Comments and Suggestions for Authors
Your article is dense, well-elaborated, and structured. The two studies it presents are well-linked and complemented the contribution of the article to pointing the impact of the meaningful learning for the well being of students.
It would be helpful to give more details about the "postgraduate students". It could be noticed that they are aged from 20-34 or more years old. What it their background and context, profile for being included in the study?
The procedure of intervention for the experimental design is well described, even a more explicit justification for the chosen Mindful Agency Coaching approach in relation with the focus of the research would be helpful. Also, in the data analysis it will be good to see if there are differences between the subjects already used with mindfulness practices and the ones being newly introduced.
Author Response
Reviewer #3:
- It would be helpful to give more details about the "postgraduate students". It could be noticed that they are aged from 20-34 or more years old. What is their background and context, profile for being included in the study?
Response:
Thank you for your feedback. We have provided additional details about the postgraduate students in our inclusion criteria on page 4: “The inclusion criteria for participants were individuals aged between 18 and 60, native Mandarin speakers, and current postgraduate students.”
- The procedure of intervention for the experimental design is well described, even a more explicit justification for the chosen Mindful Agency Coaching approach in relation with the focus of the research would be helpful.
Response:
Thank you for your suggestion. The MAC approach to develop mindful learning has been utilized and proven to be an effective model in enhancing mindful agency (please see the references 81, 82). Since we measured mindful learning through the assessment of mindful agency, the MAC intervention could be useful in the current study as well. We have included the justification on page 10: “The current MLC design was based on the original framework of Mindful Agency Coaching that can effectively develop mindful agency [81, 82], with substantial adaptations for the postgraduate students in this study.”
- Also, in the data analysis it will be good to see if there are differences between the subjects already used with mindfulness practices and the ones being newly introduced.
Response:
Thank you for your suggestion. We have incorporated your feedback into the results section on page 11, specifically addressing differences between participants with prior mindfulness experience and those newly introduced to mindfulness practices.
“In the experimental group, we conducted Mann-Whitney U tests and found no significant differences in the changes of mindful learning total score (p = .055), Psychological Well-Being (PWB) total score (p = .594), and Subjective Well-Being (SWB) total score (p = .371) between participants with and without prior meditation experience.”